# Statistical Methods in the Study of Protein Binding and Its Relationship to Drug Bioavailability in Breast Milk

**DOI:** 10.3390/molecules27113441

**Published:** 2022-05-26

**Authors:** Karolina Wanat, Elżbieta Brzezińska

**Affiliations:** Department of Analytical Chemistry, Faculty of Pharmacy, Medical University of Lodz, 90-419 Łódź, Poland; elzbieta.brzezinska@umed.lodz.pl

**Keywords:** protein binding, breast milk, M/P ratio, statistical modeling, molecular descriptors, chromatographic descriptors, affinity chromatography

## Abstract

Protein binding (PB) is indicated as the factor most severely limiting distribution in the organism, reducing the bioavailability of the drug, but also minimizing the penetration of xenobiotics into the fetus or the body of a breastfed child. Therefore, PB is an important aspect to be analyzed and monitored in the design of new drug substances. In this paper, several statistical analyses have been introduced to find the relationship between protein binding and the amount of drug in breast milk and to select molecular descriptors responsible for both pharmacokinetic phenomena. Along with descriptors related to the physicochemical properties of drugs, chromatographic descriptors from TLC and HPLC experiments were also used. Both methods used modification of the stationary phase, using bovine serum albumin (BSA) in TLC and human serum albumin (HSA) in HPLC. The use of the chromatographic data in the protein binding study was found to be positive —the most effective application of normal-phase TLC and HPLC_HSA_ data was found. Statistical analyses also confirmed the prognostic value of affinity chromatography data and protein binding itself as the most important parameters in predicting drug excretion into breast milk.

## 1. Introduction

Excretion of drugs into breast milk is an important aspect to be considered in the pharmacotherapy of breastfeeding women. Due to ethical considerations, in vivo studies are very rare and it is difficult to obtain the milk-to-plasma (M/P) ratio of many active pharmaceutical compounds (APIs). A mathematical model capable of calculating M/P values using the available data will greatly facilitate the study of the bioavailability of new APIs.

In the previous articles [1,2], we presented a comparison of statistical methods in the study of drug excretion into breast milk with the use of the M/P descriptor. It was shown that the multiple linear regression (MLR) and random forest (RF) analyses were most effective in describing this pharmacokinetic phenomenon, with the use of chromatographic data and physicochemical properties of the tested compounds. These analyses did not deviate from the known principles of bioavailability to breast milk and showed a close relationship between M/P and the level of drug–protein binding (PB) as well as the state of ionization of the API in the bloodstream. 

The papers also describe the most effective conditions for thin layer chromatography (TLC) as an analytical model for predicting the penetration of drugs into breast milk. According to these results, it can be assumed that the use of drug–protein binding indices, together with chromatographic data, will make it possible to predict the level of drug distribution into breast milk.

The main aim of this study is to provide supplementary analyses, which include: determination of physicochemical parameters related to drug protein binding; searching for a mathematical model of PB and/or M/P prediction; and the use of affinity chromatography data as an index of pharmacokinetic properties. 

The goal of developing such a model is its further utility in predicting the PB of newly developed active pharmaceutical ingredients. Only easily available API properties are needed to use the model. It can facilitate the process of introducing a new drug to use and reduce expensive in vivo testing. 

In this study the following statistical methods were used: cluster analysis (CA), discriminant function analysis (DFA) and principal component analysis (PCA) random forest regression (RF). All molecular descriptors used in this study are listed and described in Table 1.

## 2. Results

### 2.1. Correlation Analyses

The experiment investigated the results of using data from several chromatographic analysis experiments (HPLC_HSA_, NP TLC, RP TLC and, additionally, HPLC_IAM_) in predicting drug binding to protein, and thus bioavailability to breast milk. A group of 165 APIs was analyzed, in which acidic, basic and neutral drugs were observed. The best correlation with PB values was shown in the results of the HPLC_HSA_ and NP TLC experiments, in the form of log k and R_f_ values, (HPLC_HSA_: *n* = 165, R = 0.39); (NP TLC: *n* = 162, R = 0.31). The relationship is directly proportional. This is the result for all kinds of relationships. Much better results were obtained for acidic drugs (R = 0.50), even considering the smaller number of cases (*n* = 34) (Table A1, Appendix B).

Then the effect of the most frequently mentioned molecular descriptors, related to drug distribution into breast milk and protein binding, was investigated. In all groups of APIs, molecular descriptors related to the hydro-lipophilic nature of drugs play a dominant role. The most important parameters are the partition coefficient and the distribution coefficient (log P and log D). The ability to form hydrogen bonds (HD, HA) is visible here and the correlation with PB is significant. The ratio of neutral to dissociated form (log U/D), dissociation constant (pKa), ionization capacity of compounds (eH-eL) and other electron descriptors: eL and eH, show no significance. The influence of hydrophobic parameters (Sa, V, MW) is visible only in the form of the surface area to volume ratio (Sa/V). As can be seen above, this factor correlates inversely with all types of cases (Table A2, Appendix B).

### 2.2. Discriminant Function Analysis

All of the descriptors most strongly related to the variability of the PB, which at the same time did not limit the number of cases studied, were introduced into the discriminant function analysis (DFA). All cases were tested using the a/b/*n* code.

In the stepwise DFA, the discriminant variables included 9 out of 16 entered variables: PhCharge, B2, pKa, M/P, log k_HSA_, log k_IAM_, NP, eL and log U/D (Table 2).

The PC1 factor discriminates the groups of APIs the most (PC 1 eigenvalue = 3.61). The variables PhCharge and pKa have the most important share in its value. The PC2 factor (PC2 eigenvalue = 0.81) was shaped by the chromatographic descriptors and the ability to ionize (log U/D). The means of the canonical variables (PC1) for group a = −3.52, for group *n* = 0.03 and for group b = 2.08, therefore PC1 most strongly discriminates between groups a and b. The means of the canonical variables (PC2) for group a = −0.93, for group *n* = 0.86 and for group b = −0.97. In this case, the centroids of groups a and b are almost equal, and the group of neutral compounds (*n*) is the most discriminated against (Figure 1).

### 2.3. Principal Component Analysis

PCA was performed to determine the effect of the primary descriptors on the characteristics of the drug’s ability to pass into breast milk. In order to better visualize the obtained results from the analysis, the M/P values were converted into the scale of the drug penetration into milk—M/P_code_. The values of this indicator are in the range 1–4. Code 1 corresponds to drugs with an M/P value <0.40—completely safe; 2 corresponds to the range of 0.40–0.80—at the safety limit; 3 range 0.81–1.20—possibly over the safety limit; and 4 is M/P >1.20—dangerous. 

In the course of the analysis, the smallest number of principal components explaining the maximum range of the total variance in the group was initially established. Five factors explain 100% of the variability in the levels of drug excretion into breast milk. The first two factors, PC1 and PC2 (principal components), are described by all used descriptors. As a result, two main components explaining a total of 72% of the variability were obtained. The HPLC_HSA_, HPLC_IAM_, NP TLC and RP TLC chromatographic data is responsible for the first component, PC1 (43.26%), the second component, PC2 (28.66%), is determined by the PB value.

The projection of cases on the PC1 × PC2 plane is presented below (Figure 2):

In the graph of the projection of cases onto the PC plane, where the grouping variable is the scale of drug penetration into breast milk (M/P_code_), it can be seen that the tested APIs can be divided into two groups (surrounded by a box in the graph). One group included drugs with a lower level of M/P (1–2) penetration—safe, and the other group, M/P 3–4—dangerous. This division is not entirely obvious. It was created on the basis of factors explaining 75% of the variability. Few examples of misclassification are visible. The distinction between these groups is related to PC1. Derivatives with a low M/P are located on the right side of the plot and are clearly related to the positive values of PC1. APIs easily excreted into milk are on the left side of the chart and have negative PC1 values. The share of variables in this component, determined by the PC1-variable correlation (factor loadings), reveals the parameters of the greatest importance for the investigated pharmacokinetic feature of drugs. They are: log k_HSA_, log k_IAM_, NP and RP. Thus, affinity chromatography, based on protein binding, can predict the bioavailability of an API into breast milk.

The graph of the projection of variables onto the PC plane shows graphically the relationship between the component and the variable. The graph shows the so-called unit circle, i.e., the maximum correlation of 1 between the variable and the factor. The closer a given variable is to the unit circle line, the greater its correlation with the observed phenomenon (Figure 3).

### 2.4. Cluster Analysis

In order to emphasize the diagnostic value of the experiment and to determine the difference in the values of the parameters determining the ability of drugs to penetrate into breast milk, cluster analysis (CA) was also performed. CA was conducted in the proposed M/P_code_ scale, using the k-means method. The means of the most important biological descriptors (CNS +/−, B1, PhCharge, acid/base, NP, RP, log k_HSA_, log k_IAM_ and PBcode) were compared for groups M/P_code_ 1–4. As shown, all drug biological parameters showed a group variability (see Figure 4). The M/P code values range from 1 to 4 with a clear distinction between relatively safe and unsafe groups. Physicochemical parameters: PB, acid/base, HD, log P, eL, log D also show differentiation, but not in all cases. Unfortunately, M/P_code_ is too clustered here, which indicates a smaller influence of the tested properties on the observed feature (Figure 5). The descriptors: log D and eL show the highest differentiation.

The above analyses confirmed the values of the parameters HA, log P, log D and eL. The parameters of log D, HA and eL show the greatest differentiation. Unfortunately, the M/P_code_ values are poorly differentiated and their values do not correspond to the variability of other descriptors. 

### 2.5. Regression Methods

MLR failed to create a reliable PB prediction model, therefore an attempt was made to analyze protein binding by other regression methods. A total of 165 test compounds and 22–23 independent variables were used to perform partial least squares (PLS) and random forest regression (RF). The variables used are listed for each model (Table 3 and Table 4). During the analyses, 165 compounds were randomly divided into a training set, 70% of the total (TRAIN, *n* = 115 compounds,) and a test set for external validation, 30% of the total (TEST, *n* = 50).

#### 2.5.1. Partial Least Squares Regression

The PLS model using 23 independent variables, including NP TLC data (Table 3) showed low values of R^2^ and Q^2^, approximately 0.40, and even lower results of external validation, approximately 0.22–0.24 (Figure A1, Appendix B). Even lower values are achieved with the HPLC_HSA_ chromatographic data. This indicates that, as in the case of breast milk prediction models, the PLS method is again not widely applicable here and is not an appropriate method to analyze this type of data.

#### 2.5.2. Random Forest Regression

RF regression was performed with the use of 150 generated random trees. NP TLC data was used first. The independent variables used for the analysis of all 165 cases (independent variable, PB_abn_) are listed in Table 3. 

The obtained model (Figure 6) showed satisfactory results, especially for the training set (*n* = 115): R^2^_train_ = 0.81; Q^2^_train_ = 0.73. The results of external validation using the test kit (n_abn_ = 50) were lower: R^2^_test_ = 0.65; Q^2^_test_ = 0.56. The Monte Carlo permutation test (MCPT) showed the average value of the Q^2^_test_ parameter was equal to 0.56 (Appendix B, Figure A2), which is similar to that in the presented model. The influence of individual independent variables on the model is presented in the chart below (Appendix B, Figure A3). The order of the descriptors presented there is as shown in Table 3. The log D parameter shows the strongest influence on the model using NP TLC data.

The data from the HPLC_HSA_ experiment were then used for the RF regression (Table 4). The obtained model (Appendix B, Figure A4) again shows good results of the training set (*n* = 115): R^2^_train_ = 0.81; Q^2^_train_ = 0.78 but much lower parameters were obtained with external validation (n_abn_ = 50): R^2^_test_ = 0.57; Q^2^_test_ = 0.53. In the MCPT, the Q^2^_test_ value was already at a low level and amounted to 0.35 (Appendix B, Figure A5).

Then, individual groups of compounds were dealt with, either separately, (a), (b) and (*n*), or combined, (an), (bn) and (ab). The results are shown in Table 5. Only the NP TLC data (Table 3) were used to construct the models, which gave the best results when tested for the complete set of compounds (n_abn_ = 165).

RF models for PB_a_ (n_a_ = 35) and PB_b_ (n_b_ = 50) gave poor results, especially in the external validation, similarly to their combined group (n_ab_ = 85), where the external validation results were in the range of Q^2^ = 0.4–0.3.

The best results were obtained for the PB_n_ (n_n_ = 82) and PB_an_ (n_an_ = 117) groups. The R^2^ and Q^2^ values of the test kits ranged between 0.55 and 0.62 (Appendix B: Figure A6 and Figure A7). In both models, the log D values are the most important in their creation (Appendix B: Figure A8 and Figure A9).

## 3. Discussion

On the basis of the DFA analysis, it was possible to determine the influence of the acidic, basic and neutral properties of APIs on their protein binding capacity and to decide whether the analysis of the pharmacotherapy of nursing mothers (M/P predictions) should be divided into groups: a, *n* and b. The division into acidic, basic and neutral drugs is strongly related to the PB-related descriptors, so the use of groups a, b and *n* seems to bring value for further analysis. The low values of Wilks lambda for both roots, PC1 and PC2, confirm the value of the obtained results (0.11 and 0.54, respectively). 

As the DFA analysis revealed a group of physicochemical and chromatographic parameters important for the bioavailability of drugs to milk, the use of CA emphasized the differentiation of their mean values in the M/P 1–4 groups. The above analyses confirmed the values of the parameters HA, log P, log D and eL. The parameters of log D, HA and eL show the greatest differentiation. Unfortunately, the M/P_code_ values are poorly differentiated and their values do not correspond to the variability of other descriptors. Based on the PCA, it can be concluded that the data of the drug–protein binding affinity chromatography, in the form of the proposed analytical models and the protein binding itself as the basis for the experimental design, are the most important parameters in predicting drug excretion into breast milk.

The final step in this study was to construct a model capable of predicting PB value, used as a trait strongly correlated with the bioavailability of breast milk. Unfortunately, it was not possible to obtain an MLR or PLS algorithm for protein binding prediction, that was reproducible for different groups. Models created by regression using the random forest method show a significant relationship, visible in the scatter plots (Figure 6, Figure A4, Figure A6 and Figure A7). The influence of the determination coefficient (log D) and chromatographic parameters from the NP TLC and HPLC_HSA_ experiments in each model are also noticeable. Unfortunately, they do not show the best predictive ability (external validation at the level of Q^2^_test_ = 0.56 and 0.35 in MCPT tests).

The best results using random forest regression were obtained for the entire set of compounds, PB_abn_, and for the PB_n_ and PB_an_ groups. It is the acidic and neutral compounds that bind primarily to albumin, which constitutes the majority of plasma proteins, so the literature values of protein binding (PB) refer mainly to the binding of drugs to HSA.

## 4. Materials and Methods

### 4.1. Molecular Descriptors

All tested drugs are listed in Appendix A, along with molecular descriptors. Active pharmaceutical ingredients were extracted from pharmaceutical formulations, purchased in a generally accessible pharmacy. The main criterion used in composing the drug set was the availability of protein binding values (PB) along with milk-to-plasma ratios for each API, as these were the main pharmacokinetic phenomena studied.

The molecular descriptors selected for statistical analyses, which should have a significant effect on the penetration into breast milk and protein binding, are listed in Table 1. Some were taken from the literature, including M/P ratio obtained in vivo [8,9,10,11,12,13] or from online databases DrugBank [7] and CHEMBL [3]. Most of the physicochemical data were calculated in the following programs: HyperChem (HyperChem for Windows version 7.02, HyperCube Inc, Gainesville, FL, USA, 2002) and ACD/Labs (ACD/LabsTM Log D Suite 8.0, pKa dB 7.0, Advanced Chemistry Development Inc., Toronto, Canada, 2004). 

Chromatographic descriptors were obtained in experiments, thin layer chromatography in normal (NP TLC) and reversed mode (RP TLC). The stationary phase was modified with bovine serum albumin (BSA). TLC was the source of retention factor (R_f_) values, denoted in statistical models as NP and RP. High performance liquid chromatography was performed using immobilized human serum albumin column (HPLC_HSA_) and immobilized artificial membrane (HPLC_IAM_). HPLC was the source of the log k values (logarithm of retention factor), log k_HSA_ and log k_IAM_. The TLC and HPLC experiments are detailed in Appendix C.

### 4.2. Statistical Analyses

DFA, PCA and CA were performed in STATISTICA 13.1 (TIBCO Software Inc., Palo Alto, CA, USA). DFA is a classification analysis determining which descriptors best define the assignment of individual cases to each of the predetermined groups. Wilks’ lambda is a parameter used to evaluate the discriminant power of the entire model, i.e., all the independent variables used, and takes values from 0 to 1; the closer these values are to zero, the more discriminatory the model becomes. 

PCA is used to combine highly correlated variables with one another into one new variable called the principal component (PC). The calculation of new factors consists in diagonalizing the correlation or covariance matrix. The choice of matrix depends on whether the original variables require standardization or centering to mean values. In this way, a reduced number of new variables is generated, but explaining the original variance as much as possible.

The purpose of cluster analysis (CA) is to combine cases into groups so that the association within the same group is as large as possible, and with cases from other groups as small as possible. The method of grouping the data used in the presented studies was the k-means method, in which the means for each cluster and in each dimension are examined, which allows assessment of to what extent the created clusters are different from each other. In the analysis of variance, the size of the F statistic performed in each of them shows how well a given dimension separates individual clusters. In the best situation, very different means are obtained for most of the dimensions analyzed.

PLS and RF regression were performed with MATLAB ver. 2019a (The MathWorks, Natick, MA, USA). The performance of the models was assessed by a double cross-validation. The statistical significance was then evaluated using permutation testing. 

In the PLS method, the matrix of independent variables is analyzed for latent variables (LVs) that best describe the covariance between X and Y. Then these transformed independent variables are used in regression to predict the Y response. The RF method uses many decision trees which, based on the entered X variables, repeatedly “make a decision” about the predicted value of Y for each case, from which the mean value is then taken. 

In regression analyses, it is good practice to divide the set of cases into two sets: training and testing, in order to perform external validation, which will demonstrate the predictive capacity of the model. The training set accounts for approximately 70% of all collected cases and is used to build a regression equation (training model). The rest, i.e., about 30% of cases, are included in the test set on which the equation is validated. The training and test sets are distributed randomly. In order to check the stability of the model and exclude random effects, it is worth carrying out such a division into two subsets and the construction of the equation several times. The Monte Carlo permutation test (MCPT) is used for this. For the training and test sets, RF regression was performed and RMSECV, Q^2^ and R^2^ were calculated. Then this procedure was repeated 100 times, each time the training and test sets were drawn anew. Furthermore, the distribution of Q^2^ in the original and permuted models was compared and a one-way ANOVA was performed. In the next step, 100 training (70%) and test sets (30%) were prepared by randomly splitting the original data matrix. A similar MCPT (100 perm.) was then performed on the training and test sets that were derived from the permuted data matrix. The results of the original and permuted models were obtained and their Q^2^ values were compared. 

## 5. Conclusions

Positive results were obtained on the expediency of using chromatographic data in the study of protein binding and the penetration of drugs into breast milk. The presented statistical analyses showed a close relationship between HPLC and TLC analytical data (under set conditions) with the bioavailability of the drug into breast milk. The correlation of the PB and M/P ratios with these chromatographic data is high, also in the group of all cases (acidic, basic and neutral drugs) together. The most effective application of NP TLC and HPLC_HSA_ data was found. There is also a greater correlation between PB and the chromatographic data in the group of acidic drugs (a), i.e., for specific binding to albumin.

The PCA and DFA analyses identified a group of physicochemical and chromatographic parameters important for the bioavailability of drugs in breast milk. The use of CA emphasized the differentiation of their mean values in groups M/P_code_ 1–4.

NP TLC was proved to be the most useful chromatographic method in statistical analyses. In the case of HPLC_HSA_ data, the relatively large share of the results from the column in the creation of the RF model turned out to be interesting. The second factor that emerges in almost all analyses is the high proportion of the log D parameter, i.e., lipophilicity associated with ionization.

## Figures and Tables

**Figure 1 molecules-27-03441-f001:**
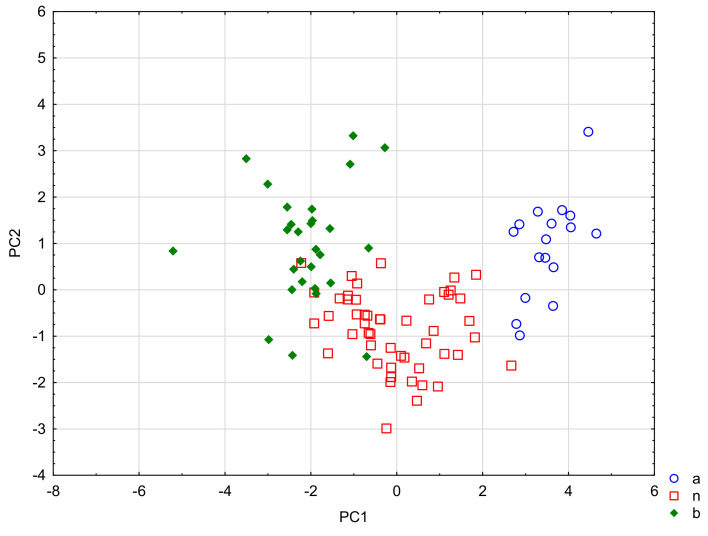
Discrimination against acidic (**a**), basic (**b**) and neutral drugs (*n*). The scatter plot of canonical values for root 1 relative to root 2. Discriminating variables: PhCharge, B2, pKa, M/P, log k_HSA_, log k_IAM_, NP, eL, log U/D.

**Figure 2 molecules-27-03441-f002:**
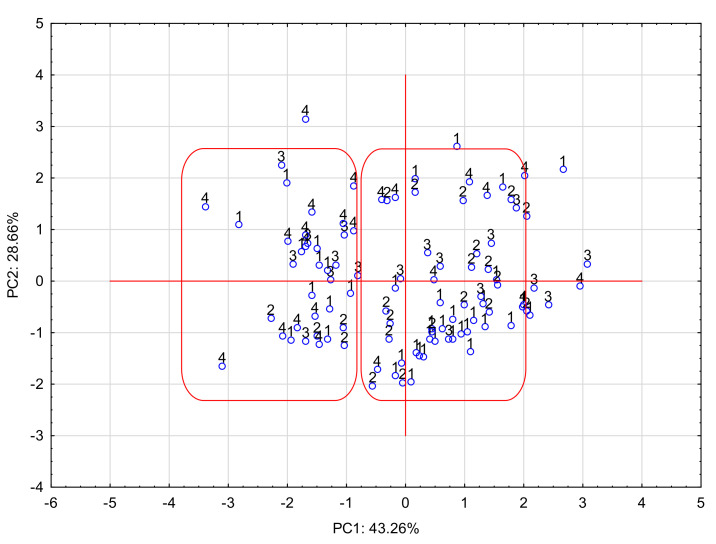
Projection of cases onto the PC1 × PC2 plane.

**Figure 3 molecules-27-03441-f003:**
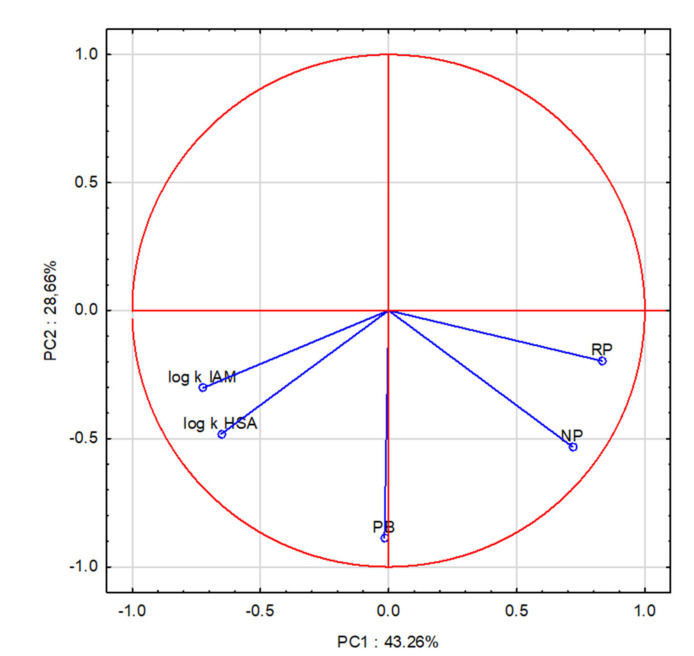
Projection of variables on the plane of factors PC1 × PC2.

**Figure 4 molecules-27-03441-f004:**
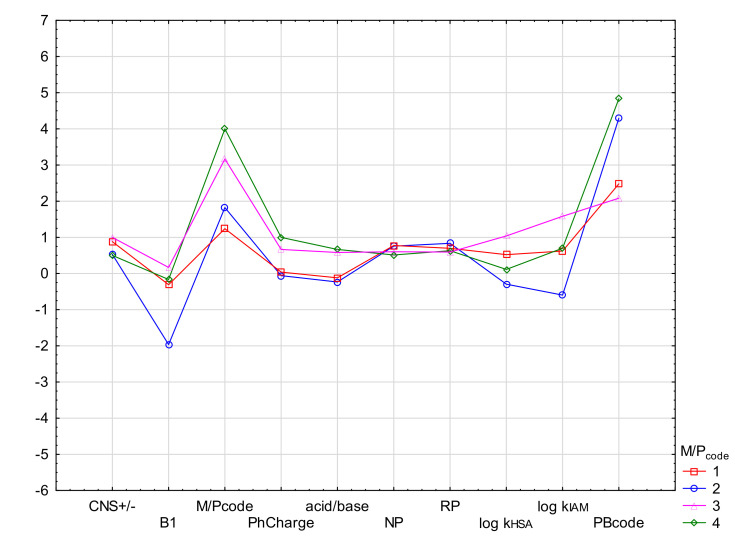
Mean descriptor values in M/P_code_ cluster analysis (k-means method) using biological and chromatographic descriptors.

**Figure 5 molecules-27-03441-f005:**
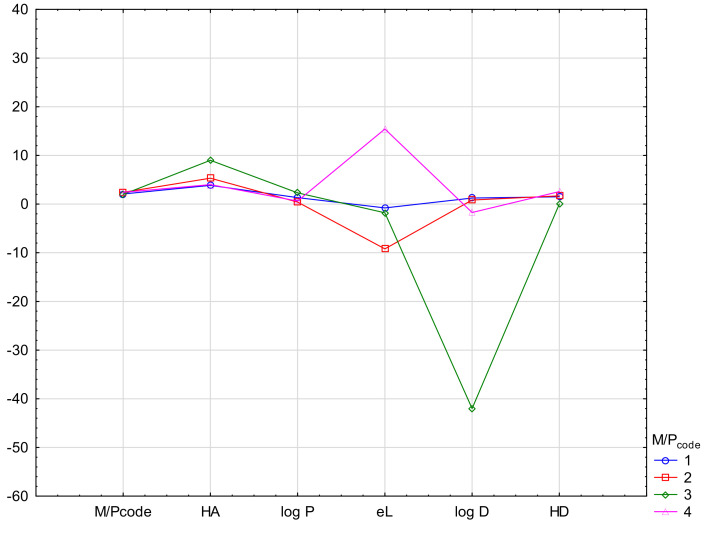
Mean descriptor values in M/P_code_ cluster analysis (k-means method) using physicochemical descriptors.

**Figure 6 molecules-27-03441-f006:**
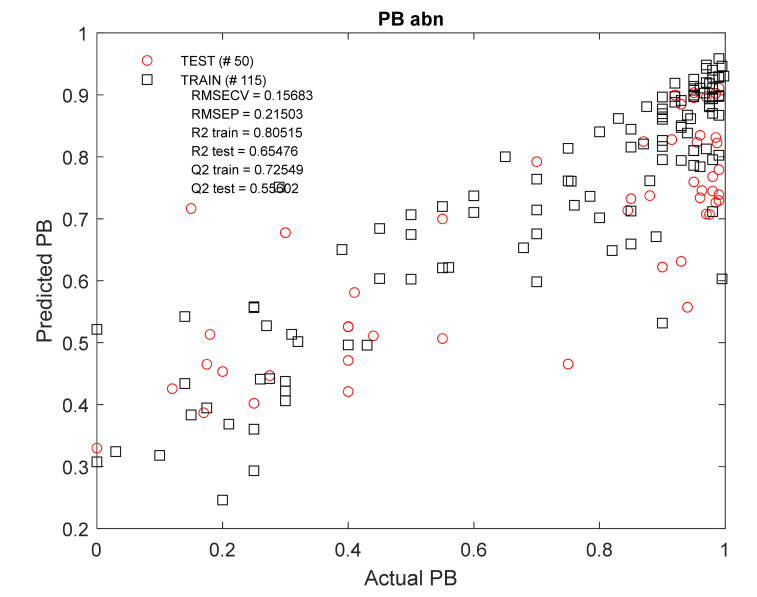
Actual versus predicted PB_abn_ values using RF regression modelling of molecular descriptor set containing 23 variables. RMSE_CV_ = root-mean-square error of cross-validation, RMSE_P_ = root-mean-square error of prediction, R^2^ train/test = coefficient of determination for train/test set models, Q^2^ train/test = coefficient of determination for the cross-validated models.

**Table 1 molecules-27-03441-t001:** List of molecular and chromatographic descriptors used in statistical analyses.

Descriptor	Description	Reference/Database/Software
a/b/*n* code	acidic, basic or neutral character of the compound; describes the division into groups: a, b and n	CHEMBL database [3]
B1	calculation parameter B2, describes the bioavailability in the CNS and determines penetration through the blood-brain barrier: log bb = 0.139 + 0.152 log P	reference [4]
B2	calculation parameter B2, describes the bioavailability in the CNS and determines penetration through the blood-brain barrier: log bb = 0.547 − 0.016 PSA	reference [5]
B3	calculation parameter related to protein binding: log (bound fraction/unbound fraction) = 0.5 log P–0.665	reference [6]
CNS+/−	ability to penetrate into the central nervous system (+ or −)	DrugBank database [7]
DM	dipole moment	HyperChem, Hypercube, Inc.
eH	energy of the highest occupied molecular orbital (HOMO)	HyperChem, Hypercube, Inc.
eH-eL	ionization capacity	HyperChem, Hypercube, Inc.
eL	energy of the lowest unoccupied molecular orbital (LUMO)	HyperChem, Hypercube, Inc.
HA	number of hydrogen bond acceptors	ACD/Labs
HD	number of hydrogen bond donors	ACD/Labs
log D	distribution coefficient	ACD/Labs
log M/P	logarithm of M/P	
log MW	logarithm of MW	
log P	partition coefficient	HyperChem, Hypercube, Inc.
log U/D	the ratio of neutral to ionized form; determines the degree of ionization	Calculated using: pK_a_-pH for acids; pH-pK_a_ for bases
M/P	milk/plasma drug concentration ratio	references [8,9,10,11,12,13]
MW	molecular weight	HyperChem, Hypercube, Inc.
PB	percentage of plasma protein binding	DrugBank
PhCharge	the charge of the API under physiological conditions	DrugBank
pK_a_	negative logarithm of the acid dissociation constant (K_a_)	ACD/Labs
PSA	polar surface area	ACD/Labs
Sa	the surface area of the molecule	HyperChem, Hypercube, Inc.
V	the volume of the molecule	HyperChem, Hypercube, Inc.
NP; RP	R_f_ (retention factor) obtained from TLC using impregnated with bovine serum albumin (BSA) plates in normal and reversed phase	TLC experiment
NP/C; RP/C	R_f_ from impregnated NP or RP plate/control R_f_	TLC experiment
k_HSA_	retention factor from HPLC using column with immobilized human serum albumin (HSA)	HPLC experiment
log k_HSA_	logarithm of the retention coefficient obtained from HPLC_HSA_	HPLC experiment
log k_IAM_	logarithm of the retention coefficient obtained from HPLC_IAM_ (column with immobilized artificial membrane)	HPLC experiment

**Table 2 molecules-27-03441-t002:** Classification matrix for the model using discriminant variables: PhCharge, B2, pKa, M/P, log k_HSA_, log k_IAM_, NP, eL, log U/D.

API Group	Correctly Classified Cases (%)	a *p* = 0.17895	*n* *p* = 0.52632	b *p* = 0.29474
a	100,00	17	0	0
*n*	96,00	0	48	2
b	92,86	0	2	26
all	95,80	17	50	28

**Table 3 molecules-27-03441-t003:** Twenty-three independent variables with NP TLC data used to create the RF and PLS model for PB.

No.	Independent Variable	No.	Independent Variable	No.	Independent Variable
1.	B3	9.	NP/B2	17.	eH
2.	PhCharge	10.	NP/log P	18.	eL
3.	acid/base	11.	MW	19.	eH-eL
4.	pKa	12.	log MW	20.	logD
5.	log U/D	13.	PSA	21.	Sa
6.	C	14.	HD	22.	V
7.	NP	15.	HA	23.	logP
8.	NP/C	16.	DM		

**Table 4 molecules-27-03441-t004:** Twenty-two independent variables with HPLC_HSA_ data used to create the RF and PLS model for PB.

No.	Independent Variable	No.	Independent Variable	No.	Independent Variable
1.	B3	9.	log k_HSA_/log P	17.	eL
2.	PhCharge	10.	MW	18.	eH-eL
3.	acid/base	11.	log MW	19.	log D
4.	pKa	12.	PSA	20.	Sa
5.	log U/D	13.	HD	21.	V
6.	k_HSA_	14.	HA	22.	log P
7.	log k_HSA_	15.	DM		
8.	log k_HSA_/B2	16.	eH		

**Table 5 molecules-27-03441-t005:** Random forest regression results on individual drug combinations.

API Group	Train Set	Test Set
PB_a_	*n* = 24 R^2^ = 0.78; Q^2^ = 0.62	*n* = 11 R^2^ = 0.29; Q^2^ = 0.11
PB_b_	*n* = 35 R^2^ = 0.88; Q^2^ = 0.80	*n* = 15 R^2^ = 0.33; Q^2^ = 0.29
PB_n_	*n* = 57 R^2^ = 0.85; Q^2^ = 0.81	*n* = 25 R^2^ = 0.62; Q^2^ = 0.59
PB_an_	*n* = 82 R^2^ = 0.82; Q^2^ = 0.74	*n* = 35 R^2^ = 0.60; Q^2^ = 0.55
PB_bn_	*n* = 92 R^2^ = 0.85; Q^2^ = 0.80	*n* = 40 R^2^ = 0.44; Q^2^ = 0.44
PB_ab_	*n* = 59 R^2^ = 0.80; Q^2^ = 0.72	*n* = 26 R^2^ = 0.38; Q^2^ = 0.33

## Data Availability

Not applicable.

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
