# Peer review of "Statistical Methods in the Study of Protein Binding and Its Relationship to Drug Bioavailability in Breast Milk"

_molecules, 2022, doi:10.3390/molecules27113441_

Round 1

Reviewer 1 Report

The manuscript by Wanat & Brzezinska “Statistical methods in the study of protein binding and its relationship to drug bioavailability in breast milk” is the continuation of their previous work (References 1 and 2) about excretion of drugs into breast milk. In this research authors are searching for new mathematical model capable of predicting Protein Binding and Milk-to-Plasma ratio using a number of theoretical and experimental (chromatographic) descriptor on the training set made of 129 known active pharmaceutical compounds.

 In my opinion research is done correctly following the modern trends in statistical analysis. The conclusions are meaningful and well supported by the presented data. Manuscript is well written, easy to follow and understand. English is good, I have found no typos or grammatical errors.

The data and conclusions presented in the manuscript can be of interest to the readers of MOLECULES journal and I recommend publication of this manuscript.

Author Response

I am glad that the manuscript has been positively reviewed.

Reviewer 2 Report

This manuscript describes statistical methods in the study of protein binding and its relationship to drug exposure in breast milk. The following comments should be properly addressed prior to further consideration.

1. The precision and accuracy of the model prediction should be validated by estimating relevant parameters.

2. The clinical relevance and applicability of this model should be justified. For what purpose can this model be used?

3. Provide more details of the 167 drugs tested in this study, together with their inclusion criterion, if any.

Author Response

Thank you for reading the manuscript and for pointing out issues that require further clarification. Below, I explain the doubts that have been reported.

  1. The precision and accuracy of the model prediction should be validated by estimating relevant parameters. 

The random forest prediction models have been validated both internally by cross-validation and externally, providing a test set (30% of drugs) that is not used in model development. Validation was expressed using the following parameters: RMSECV = root-mean-square error of cross-validation, RMSEP = root-mean-square error of prediction, Q2 train/test = coefficient of determination for the cross-validated models; everything was described in captions under the Figures. Additionally, the Monte Carlo Permutation Test  (MCPT) was provided for each model, in order to check the stability of the model and exclude random effects. MCTP procedure has been explained in details in the Materials and methods section.

  1. The clinical relevance and applicability of this model should be justified. For what purpose can this model be used?

The random forest regression model was build using drugs with known protein binding values (PB). The main goal of developing such a model is its further utility in predicting the PB of newly developed active pharmaceutical ingredients (APIs). Only easily available API properties are needed to use the model. It can facilitate the process of introducing a new drug to use and reduce expensive in vivo testing.

 This annotation was added to the revised version of the manuscript (in the Introduction section, 5th paragraph).

  1. Provide more details of the 167 drugs tested in this study, together with their inclusion criterion, if any.

 All collected drugs are commonly used to treat various ailments. Active pharmaceutical ingredients (APIs) were extracted from pharmaceutical formulations, purchased in a generally accessible pharmacy. The main criterion used in composing drug set was the availability of protein binding values (PB) along with milk-to-plasma ratios for each API, as these were the main pharmacokinetic phenomena studied.

This annotation was also added to the revised version of the manuscript (in the Materials and methods section: Molecular descriptors, 1st paragraph).

Round 2

Reviewer 2 Report

The revised manuscript is now acceptable for publication in the journal.